# Out of Africa: The genomic footprints of Vietnamese Robusta coffee

Tram Vi[1,2]*, Thi Nhu Le[2], Philippe Cubry[1], Viet Ha Phan[3], Thi Tieu Oanh Dinh[3], Thi Bich Ngoc Tran[3], Van Toan Nguyen[2], Claude Patrick Millet[1], Jean-Léon Kambale[4], Gba Kossia Manzan Karine[5], Pascal Musoli[6], Ucu Sumirat[7], Jose Cassule Mahinga[8], Piet Stoffelen[9], Dapeng Zhang[10], Pierre Marraccini[1,11], Yves Vigouroux[1], Ngan Giang Khong[2°], Valerie Poncet[1°]*

1 UMR DIADE, Univ Montpellier, IRD, CIRAD, Montpellier, France, 2 National Key Laboratory for Plant Cellular Biotechnology, Agricultural Genetics Institute (AGI), Hanoi, Vietnam, 3 Western Highlands Agriculture & Forestry Science Institute (WASI), Buon Ma Thuot, Vietnam, 4 University of Kisangani, Kisangani, Democratic Republic of the Congo, 5 National Agronomic Research Center (CNRA), Abidjan, Côte d'Ivoire, 6 National Agricultural Research Organization (NARO), Mukono, Uganda, 7 Starbucks Farmer Support Center, North Sumatra, Indonesia, 8 National Coffee Institute (INCA), Luanda, Angola, 9 Meise Botanic Garden, Meise, Belgium, 10 USDA-ARS, SPCL, Beltsville, Maryland, United States of America, 11 CIRAD, Montpellier, France

☙ These authors contributed equally to this work.
* vbt576@gmail.com (TV); valerie.poncet@ird.fr (VP)

## Abstract

Vietnam is the main producer of Robusta (*Coffea canephora*) coffee, but faces several future agronomic challenges. These may be addressed through breeding for improved cultivars and more sustainable cropping systems. For such efforts to be successful and efficient, locally available genetic resources must be understood. Indeed, while *C. canephora* exhibits high genetic diversity in its native tropical African forests, only a part of it contributed to the worldwide diffusion of Robusta. Here we traced the African origins of Robusta accessions cultivated in the Central Highlands of Vietnam. A total of 126 Robusta accessions from the Vietnam coffee germplasm collection were characterized, including historical, elite and local cultivated clones. Their genetic diversity and origins were inferred through comparisons with wild reference samples using a new set of 261 genome-wide SNPs. A core set of 45 accessions that maximize the genetic distance and allelic richness were identified for conservation and breeding priorities. Full genome sequencing of these individuals helped to closely trace the origins of chromosomal segments back to different, geographically-structured wild African genetic groups. All Vietnamese Robusta accessions displayed Congo Basin (ER group) origins, albeit to various extents. However, we also uncovered contribution from several other genetic groups, variously from the Guinean region (D), the central African Atlantic coast (AG), and Eastern CAR/Uganda (OB), in 31 hybrid individuals. These source groups have been widely used in crossbreeding to develop elite clones. In addition, using whole-genome sequencing data, we also identified various

**Data availability statement:** The raw sequencing data of the African accessions were obtained from the NCBI SRA database under project accession number PRJNA803612 (Tournebize et al. 2022). The raw sequencing data of the 45 Vietnamese core accessions are available in the NCBI SRA database under project accession number PRJNA950219. The passport and genotyping data at 261 SNPs are available in the Supplementary data.

**Funding:** This research was funded by the Ministry of Science and Technology of Vietnam (MOST) under grant number NVQG-2020/ĐT.15. TV was funded by Ph.D. grants from the French Embassy in Vietnam and the ARTS program (IRD).

**Competing interests:** The authors have declared that no competing interests exist.

admixture patterns at the chromosome level among the hybrids, which might provide valuable information for selecting breeding materials.

## 1. Introduction

Coffee is a paramount beverage that is consumed worldwide, while also having an important environmental, economic and cultural role in many countries [1,2]. Arabica and Robusta are two main types of coffee that are produced from different species, *Coffea arabica* L. and *Coffea canephora* Pierre ex A.Froehner, respectively [3]. Robusta accounts for about 40% of global coffee production, in large part due to its use in instant coffee [4].

*C. canephora* is a diploid species with the widest native distribution of all species in the *Coffea* genus [3]. Previous studies have revealed its genetic diversity to be structured into geographically-constrained groups, corresponding mainly to different West, Central and East African regions [5–7]. The most recent study classified wild *C. canephora* in the following eight genetic groups (Fig 1A): D group (also known as the Guinean group) from Guinea and Côte D'Ivoire, A group (also known as Congolese subgroup 1 (SG1), while including the '(Petit) Kwilu' or Kouilou cultivar from Gabon), B group from the southern Central African Republic (CAR), C group from Cameroon, E group (also known as Congolese subgroup 2, SG2) from the Democratic Republic of the Congo (DRC), O group from Uganda, G group from Angola, and the R group from the Sankuru region of DRC [7–9]. However, the genetic differentiation between samples of some groups was sometimes found to be low, e.g., between O and B, E and R, or A and G groups, which are often pooled [7,10,11].

While most modern crops have undergone thousands of years of domestication since the Neolithic revolution [12], widespread *C. canephora* cultivation is a very recent phenomenon [13–15]. *C. canephora* was initially cultivated at a small scale in the late 19th century in Gabon, Angola, Uganda, and in the Sankuru region of DRC [16–19]. A first large-scale breeding program was set up in Java, Indonesia, in the 1900s [20], with the aim to replace disease-stricken *C. arabica* fields [21]. Subsequent breeding centers were set up in Africa from the 1930s until 1960: INEAC Yangambi in the DRC [22], and from 1970 until the 1980s in Côte d'Ivoire and Madagascar [18]. These breeding programs were mostly based on parental accessions from three main groups, i.e., E, A and D, used to produce elite varieties and disseminate improved materials to other parts of the world [15,18]. Cultivated Robusta populations therefore likely originated from a limited number of diversity sources and they likely experienced less drastic domestication bottlenecks and selection than other domesticated crops [23]. Still, many germplasm banks were found to have narrow genetic diversity compared to that found in the wild, and high-quality elite varieties were mostly developed from limited parental materials, mainly through crosses between Guinean and Congolese groups [18,24–28]. The recent and limited breeding sources make it easier to trace back hybrid origins at the chromosome level. Recently, a local ancestry inference (LAI) approach to detect introgression

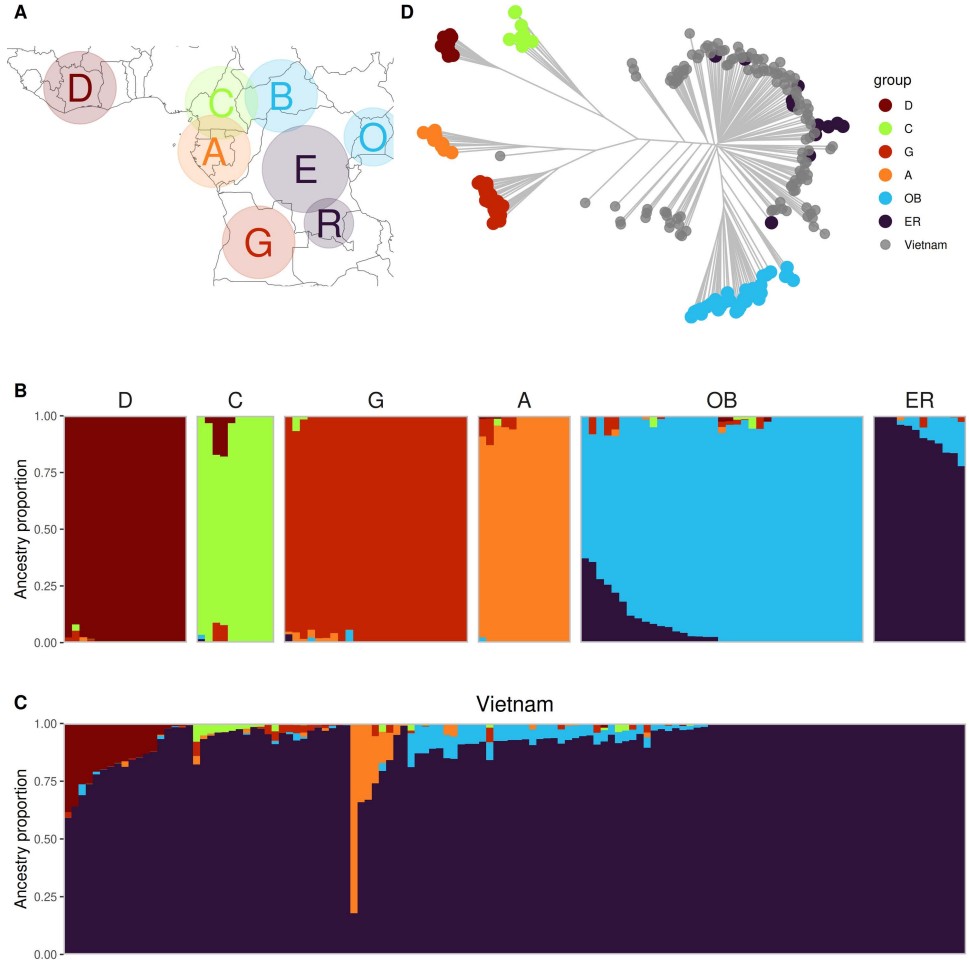

**Fig 1. Genetic origin, diversity and structure.** (A) Geographical distribution of the wild genetic groups in West and Central Africa (adapted from Mérot-L'Anthoëne et al. [7]). Genetic structure of all individuals, (B) for 127 African individuals and (C) for 126 Vietnamese individuals. The ancestry proportions of all individuals were determined via sNMF analysis with K = 6 using 261 SNPs. (D) Neighbor-joining clustering of all individuals based on the Euclidean distance.

locally at the chromosome level using single nucleotide polymorphisms (SNP) markers has been adapted for cultivated *C. canephora* [11], which helps better understand the genetic diversity and breeding history of this species.

Studying the genetic diversity of different plant resources, including cultivated and wild varieties, landraces, and germplasm collections containing elite and/or mutant varieties is crucial for crop improvement [29]. Wild crop relatives (WCRs) may bear beneficial alleles that are missing in cultivated populations, and thus in recent decades have been commonly used in breeding programs to enhance crop performance and adaptation [30–33]. For instance, genes involved in fungal blast (*Magnaporthe oryzae*) and bacterial leaf blight (*Xanthomonas oryzae*) resistance, and drought tolerance were identified in wild rice (*Oryza sativa*) and successfully introgressed into cultivar lines [34–37]. The different wild *C. canephora* groups also host genetic resources of great potential for coffee improvement. These groups have high genetic differentiation and may include variants adapted to a broad range of environmental conditions due to their wide native range [7,10]. In addition, wild coffee relatives can be screened for candidate loci and genes, such as those related to biotic stress (e.g., leaf rust [38]) and abiotic stress (e.g., drought, heat and salt [39–41]), which may be useful for enhancing coffee resilience. They can also vary in their agronomic traits across geographical locations and genetic groups [42]. For example, E

group accessions from the Congo Basin were found to feature high productivity, while A group accessions from Gabon or D group accessions from the Guinean region had high resistance to leaf rust and drought.

*Ex situ* collections of cultivated and wild germplasm are an important source of materials for commercial breeding programs [29,43]. More than 50 *ex situ* Robusta collections, containing at least 30,000 accessions, are currently hosted in around 40 countries, with the CNRA collection in Côte d'Ivoire being the largest and most representative to our knowledge [5,6,8,33,44–46]. Genetic characterization of accessions in these (inter)national collections has expanded knowledge on the extent of diversity at the regional level, but they only provide a partial view. Large collections may contain redundant individuals and their diversity may not be fully utilized, while their management is not always efficient [47]. The representative core collection concept was therefore established to choose individuals representing the genetic diversity of larger populations with minimum redundancy [48–50]. In addition, core collections are useful for identifying elite lines, beneficial genes and quantitative trait loci, as applied to many crops such as rice, soybean and cucumber [51]. Core collection sampling can be based on one or multiple features, including passport, phenotypic and genotypic data [47,52], as well as various strategies, e.g., maximizing allelic richness or phenotypic distance [53]. The setup of several Robusta core collections has been proposed based on genetic data. Gomez et al. [5] defined two core sets of CNRA accessions that capture maximum diversity using the principal component score strategy [54] on both simple sequence repeats (SSR) and restriction fragment length polymorphism (RFLP) markers. Leroy et al. [55] proposed different core collections from Robusta accessions maintained in four field collections by maximizing the genetic diversity and/or allelic diversity at 13 microsatellite markers. More recently, Verleysen et al. [22], using SNPs, also developed different core collections from 730 accessions in the INERA Yangambi Coffee Collection by combining maximized genetic diversity with minimized entry-accession distance or maximized entry-to-nearest-entry distance. Core collection conservation is essential for current and future breeding programs in Africa and abroad and necessitates a good understanding of the genetic composition, diversity, and history of the accessions and varieties that comprise them.

Robusta coffee was first introduced to Vietnam in the 1900s [56], but coffee production has only been developed over the past three decades as a major export-oriented industry to make Vietnam the second largest coffee producer in the world. Currently, Vietnam is the world's biggest Robusta producer (accounting for 40% of global Robusta production), with a cultivation area of > 600,000 ha concentrated in the Central Highlands [56]. Robusta coffee is one of the country's major cash crops, accounting for 10–15% of Vietnam's agricultural GDP [56]. The largest germplasm collection, with nearly 200 accessions, is located at the Western Highlands Agriculture and Forestry Science Institute (WASI) in Dak Lak Province [57]. Since the 1990s, various local Robusta varieties have been selected from local farms and used to develop elite varieties at WASI [58]. However, so far little is known about the origin and genetic diversity of this germplasm bank, or indeed of Vietnamese Robusta in general. Previous studies have genotyped Robusta trees labelled as "Vietnamese accessions" that were held in collections in Ghana [59] and Mexico [60]. Of the diversity held at WASI, only 10 elite clones have been genotyped [11], and were found to be related to the Congolese group. None of these studies aimed at characterizing Vietnamese coffee diversity at a large scale, while the WASI collection contains a large number of local accessions, including elite clones, historical accessions dating from the earlier days of Vietnamese Robusta cultivation, and trees collected from local smallholder farms. Therefore, the WASI collection is likely more representative of the diversity in Vietnamese coffee as a whole, and potentially presents more genetic diversity than previously found.

There is a risk of a decline in Vietnamese Robusta coffee production in the future. First, it is predicted that the most suitable existing coffee croplands in the Central Highlands will be reduced by > 50% by 2050 due to climate change [56,61]. The consequences of this climatic trend, such as rising temperatures, are also associated with the emergence of coffee pests and diseases [62]. Secondly, most Robusta trees in Vietnam are currently > 15 and 20 years old [56] and such trees often become unproductive when they reach 25–30 years of age [63]. Moreover, other abiotic and biotic factors also negatively affect Robusta coffee in the Central Highlands, e.g., polluted and exhausted soil due to inefficient irrigation

and intensive cultivation [56], as well as increased nematode pressure [64,65]. There is hence an urgent need to develop new Robusta plants that are better adapted to climate change, with greater resistance to biotic and abiotic stresses.

Understanding the genetic diversity of the WASI collection can provide a surer and more complete picture of Vietnamese germplasm resources, and developing core collections will enhance efficient conservation efforts and facilitate future breeding for Robusta improvement in Vietnam. In this study, 126 Robusta accessions were collected in the WASI germplasm bank, including elite varieties, historical accessions and other local accessions. Moreover, a set of wild African accessions spanning the species distribution range were used as genetic reference. Our study aimed to: (1) understand the origin and genetic diversity of Vietnamese Robusta coffee in relation to the wild diversity using a new SNP genotyping set, (2) propose a representative core collection to maximize the genetic diversity and minimize the genetic redundancy, and finally (3) detect admixture segments in the core individuals using whole-genome sequencing data so as to assess the diversity at the chromosome level. The results could highlight current germplasm varieties of potential interest for developing future breeding strategies.

## 2. Materials and methods

### 2.1. Plant material

A total of 126 Vietnamese Robusta accessions were collected at the Western Highlands Agriculture and Forestry Science Institute (WASI), including 10 elite accessions created and recognized by WASI (previously sequenced by Vi et al. [11]), 11 historical accessions which were introduced early in Vietnamese Robusta cultivation, and other accessions collected from local gardens (S1 Table). The elite accessions were designated as TR4, TR5, TR6, TR9, TR10, TR11, TR12, TR13, TR14 and TR15 clones. Some of these clones were those that have been the most favored and widely cultivated by Vietnamese coffee growers in recent years, especially TR14 and TR15 clones, which were found to feature late ripeness and adapt very well to climatic variations [58]. The historical "Gx" accessions were collected from the Dak Lak Museum in 2020 and were probably about 100 years old but no related information could be found.

A set of 127 African Robusta accessions (S2 Table) with known genetic groups, including 17 hybrids, were used as reference. They were from previous studies [7,10] or the present study.

### 2.2. KASPar genotypi ng data

A new set of SNPs developed for KASPar genotyping (S3 Table) was derived from the 8.5K SNP arrays [7] using an African reference set. In this dataset, genotypes from the 8.5K SNP array data reported in Mérot-L'Anthoëne et al. [7] and resequencing data from Tournebize et al. [10] were combined at the 8.5K SNPs [7]. Biallelic SNPs with <5% missing data, and minor allele frequencies >0.05 were retained. The remaining SNPs were thinned out within 40 kb distance using vcftools 0.1.16 [66].

The selected SNPs were submitted to LGC Biosearch Technologies for KASPar genotyping [67] of the Vietnamese accessions (excluding the sequenced elite accessions) and the new African accessions. Genomic DNA was extracted from leaves using the sbeadex™ mini plant kit. Flanking 120 bp sequences upstream and downstream of the SNPs were used for primer design.

A total of 261 SNPs were validated for KASPar genotyping of 116 WASI accessions, with 29 duplicates serving for quality control. For the remaining accessions, genotypes at the same loci were determined from their previous genotyping or sequencing data.

### 2.3. Genotyping analysis

We assessed the genetic diversity of the Vietnamese accessions and their relationship with the wild accessions via principal component analysis (PCA) and neighbor-joining clustering based on the Euclidean genetic distance of all individuals

using the 261 SNPs. Descriptive statistics on the African and Vietnamese groups, including the allelic richness, observed heterozygosity, expected heterozygosity and inbreeding coefficient, were computed using the R packages *adegenet* [68,69] and *hierfstat* [70]. The genetic structure was analyzed by performing sparse non-negative matrix factorization (*sNMF* [71]) for K ranging from 1 to 10 with 100 iterations. All the analyses were performed in R [72].

### 2.4. Core collection

To obtain an optimal representativeness as well as eliminate redundancy of the Vietnamese collection, we selected a representative set consisting of 45 individuals, using the R package *Core Hunter 3* [53]. As 10 accessions are recognized as elite varieties and are widely cropped in Vietnam [58] they were manually included in the core set. The selection of the remaining 35 individuals was based on maximizing the Euclidean genetic distance and allelic diversity in the core set.

We assessed the core set representativeness by generating 1,000 sets of 45 random individuals from the whole collection and comparing them with the core set for expected heterozygosity and allelic richness.

### 2.5. Sequencing and SNP calling

The sequencing data included 55 African reference samples from Tournebize et al. [10], 10 elite accessions from Vi et al. [11], and 35 core individuals newly sequenced using DNBseq PE 150 (DNA extraction, library construction and sequencing performed by BGI Hong Kong). All the raw sequencing data were cleaned by trimming Q20 read bases using cutadapt 3.1 [73]. Variant calling was performed according to GATK Best Practices recommendations for germline short variant discovery using the v 1.8 reference genome [74]. Variants were then filtered by quality, depth, missing data, singletons and doubletons. Only biallelic SNPs were retained for further analysis. The SNP calling procedure and the software used are described in S1 Fig.

### 2.6. Whole-genome SNP analysis

We assessed the genetic structure of the sequenced Vietnamese and reference samples by performing sNMF [71] on a set of genome-wide SNPs, which were filtered by minor allele frequency (MAF) > 0.05 and thinned out within a distance of 5 kb. sNMF was run for K ranging from 1 to 10 with 10 iterations, and the optimal K was determined based on the cross-entropy criterion.

Admixture segments throughout the genome were assessed by inferring the local wild ancestry-of-origin at the chromosome level for each individual in the Vietnamese core set. We used the genetic groups classified by the sNMF analysis as ancestry sources and implemented the approach developed in Vi et al. [11]. Briefly, this ELAI-derived efficient local ancestry inference method, based on a two-layer hidden Markov model [75], was applied to track segments of different ancestral origins in the studied accessions in the case of multiway admixtures.

## 3. Results

### 3.1. Genetic diversity and origin

From the 8.5K array SNPs [7], 268 biallelic SNPs distributed across all of the 11 chromosomes were chosen for genotyping. These SNPs allowed us to efficiently discriminate genotypes from different sources. For the Vietnamese Robusta accessions, 116 genotypes were obtained by KASPar genotyping, and 10 genotypes were determined from re-sequencing data [11]. For the African reference samples, 127 genotypes at the 268 loci were composed of genotypes determined from KASPar genotyping, re-sequencing data [10], and genotyping data using 8.5K array data [7]. The KASPar SNP quality was assessed using 29 duplicated samples, which showed agreement (exactly replicated genotypes at a locus) in 99.5% of all of the SNPs. Finally, we determined the genotypes of all of the Vietnamese and African individuals at 261 SNPs with < 5% missing data.

We analyzed the genetic diversity of the Vietnamese Robusta collection (126 individuals) and the African groups (127 individuals) based on these 261 genome-wide SNPs. The genetic structure of all of the individuals was analyzed using sNMF with K ranging from 1 to 10. The cross-entropy criterion did not show a clear optimum K value (S2 Fig). The African groups were found to be well structured with K=6. The six groups were named on the basis of the genetic groups corresponding to their origins [7]: D group from West Africa, C group from Cameroon, G group from Angola, A group from Gabon and Benin, OB group composed of 2 previously described groups from Uganda and CAR, and the ER group composed of 2 previously described groups from DRC (Fig 1A and B). All of the Vietnamese accessions included a proportion of the merged ER group at certain levels (Fig 1C). The majority of them (97 individuals) presented >90% ancestry from the ER group, while 6 of them were composed of 12–82% of the A group, and 13 individuals with 12–38% of the D group. Ten historical accessions in the collection also had 86–100% of the ER group, with 10% or 12% of the OB group in two accessions.

The relationship between the Vietnamese cultivated individuals and the wild accessions was assessed based on their pairwise Euclidean distances (Fig 1D). African individuals from the six groups defined by sNMF were also grouped on long branches. The ER and OB groups were closely related, which was congruent with the structure analysis as they shared some ancestry proportions. Most of the Vietnamese genotypes were clustered closely together and alongside accessions originating from DRC, i.e., the ER group, while some others were closer to the remaining groups. These results were also in line with the PCA results. The first two PCs, which explained 43.6% of the total variance, also showed clustering of Vietnamese individuals with wild groups from Central Africa (S3 Fig).

The genetic diversity of the Vietnamese population and the six African groups was assessed based on some descriptive statistics (Tables 1 and 2). African individuals presenting <70% ancestry were excluded from the analysis. The mean allelic richness (AR) of the Vietnamese group was 1.62, which was higher than that of most of the wild genetic groups (AR=1.18–1.51). The observed heterozygosity ($Ho$=0.18) and expected heterozygosity ($He$=0.19) were similar to the patterns noted for the Congolese ER group, which had higher values than the other groups, and lower than those of all of the African populations ($Ho$=0.36, $He$=0.28). The inbreeding coefficient among the Vietnamese individuals was low – $F_{IS}$=0.03, while it ranged from 0.06 to 0.38 in the wild groups. However, there were three pairs of identical genotypes

**Table 1. Summary statistics of genetic diversity of the Vietnamese Robusta collection and the genetic groups in Africa.** The African groups were classified based on the sNMF results, and the individuals with > 70% ancestry were assigned to the corresponding group. N=number of individuals; AR=average allelic richness; Ho=observed heterozygosity; He=expected heterozygosity; $F_{IS}$=inbreeding coefficient.

| Statistics | D | C | G | A | OB | ER | Vietnam |
|---|---|---|---|---|---|---|---|
| N | 16 | 10 | 24 | 12 | 32 | 12 | 126 |
| AR | 1.18 | 1.45 | 1.28 | 1.31 | 1.42 | 1.51 | 1.62 |
| Ho | 0.04 | 0.11 | 0.08 | 0.1 | 0.12 | 0.17 | 0.18 |
| He | 0.05 | 0.14 | 0.08 | 0.1 | 0.15 | 0.18 | 0.19 |
| $F_{IS}$ | 0.08 | 0.19 | 0.02 | 0.06 | 0.19 | 0.06 | 0.03 |

**Table 2. Differentiation coefficient ($F_{ST}$) between the wild and Vietnamese groups.** All the $F_{ST}$ values are statistically significant (p=0.000).

| | D | C | G | A | OB | ER |
|---|---|---|---|---|---|---|
| C | 0.687 | | | | | |
| G | 0.814 | 0.679 | | | | |
| A | 0.835 | 0.705 | 0.536 | | | |
| OB | 0.753 | 0.592 | 0.677 | 0.708 | | |
| ER | 0.774 | 0.573 | 0.67 | 0.684 | 0.339 | |
| Vietnam | 0.654 | 0.515 | 0.558 | 0.612 | 0.309 | 0.035 |

in the Vietnamese collection, suggesting putative clones or possible mislabeling. The $F_{ST}$ differentiation coefficients were also high between the African groups, ranging from 0.339 (OB and ER) to 0.814 (D and G). The $F_{ST}$ values between the Vietnamese group and most of the wild groups were high, ranging from 0.309 (group OB) to 0.654 (group D), except for the ER group, with $F_{ST}=0.035$, which further confirmed their close relationship.

### 3.2. Core set construction

As the Vietnamese Robusta genotypes had a high redundancy level, we selected a core set containing the most representative individuals, i.e., maximizing the genetic distance and allelic richness. The core set of 45 accessions had markedly higher expected heterozygosity (He=0.22) and allelic richness (AR=1.91) than randomly selected sets of the same size (Fig 2A and B), while being comparable to the values of the whole collection (He=0.21, and AR=1.41). Most of the closely related individuals were removed from the core set, while most of the hybrids were retained, especially highly divergent individuals which contributed to the greatest pairwise genetic distances (Fig 2C and D). These accessions were subsequently sequenced to analyse the whole genome structure in greater depth.

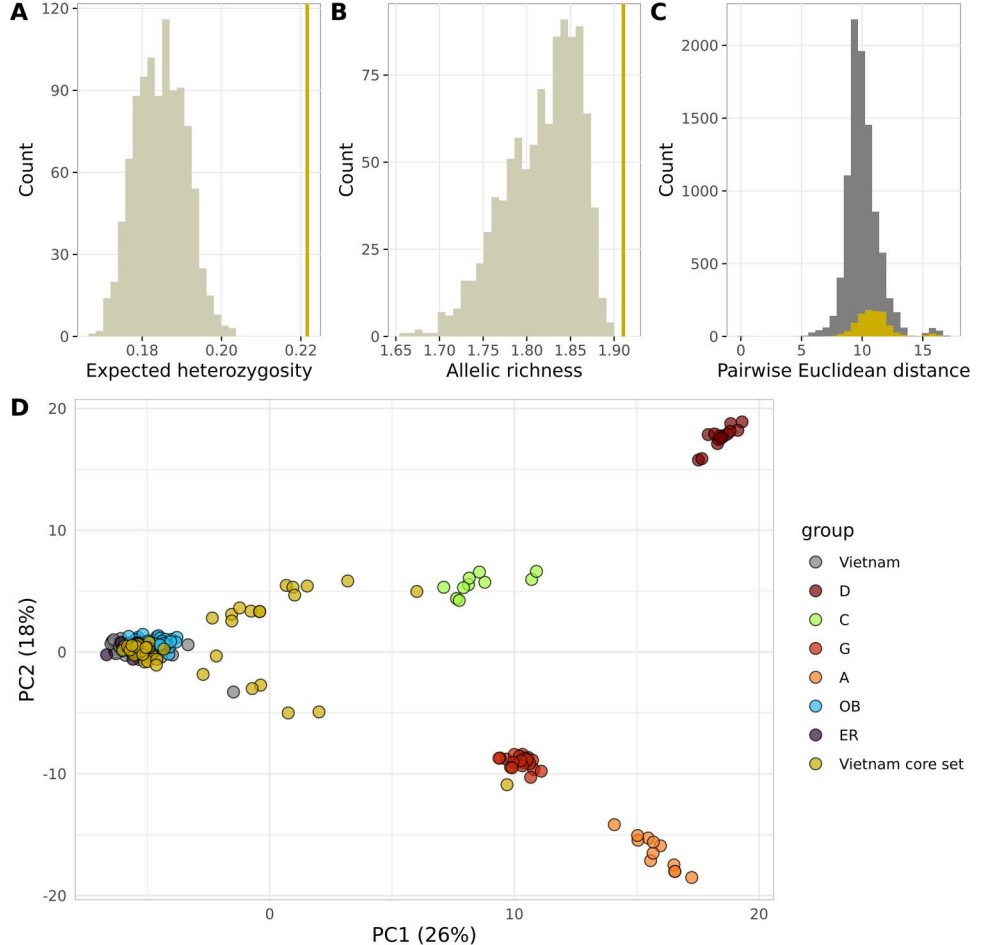

**Fig 2. Selection of a core set from the Vietnamese collection.** Evaluation of the core set: Histogram of (A) the expected heterozygosity and (B) allelic richness of 1,000 random sets of the same size as the core set (45 individuals). (C) Histogram of pairwise Euclidean distances between individuals in the whole collection (gray bars) and in the core set (yellow bars). (D) Individuals selected for the core set are highlighted (yellow) in a PCA projection.

### 3.3. Local ancestry analysis of the core set accessions

Because of the strict out-crossing nature of Robusta, admixture patterns can be increasingly complex and shortened after each successive breeding generation, which might be difficult to be detected using low-density SNP data. Therefore, to understand better and more precisely the admixture patterns of these hybrids in the core set, we detected composition of genetic groups along each chromosome (local ancestry), using whole-genome sequencing data. A total of 13,991,298 biallelic SNPs were obtained when combining the WASI core set and the reference set of 55 African wild accession sequences [10]. These SNPs were used for local ancestry inference at the whole-genome level. When using a subset of 5 kb thinned SNPs, sNMF differentiated the African reference set into five genetic groups (S4 Fig), similar to the pattern described above in section 3.1 except that the closely related A and G groups were merged due to the smaller sample size. The ancestry of the Vietnamese core individuals at the chromosome level was thus inferred from the five ancestral groups (D, C, AG, OB, and ER) that serve as source populations.

The total ancestry proportion associated with the local ancestry inference per individual was congruent with the genetic structure resulting from the sNMF analysis, with a correlation coefficient of r = 0.97. Only 0–14% of the genome remained undetermined due to uncertainties associated with the detection method [11]. All of the Vietnamese individuals presented chromosome segments of the ER group (Fig 3, and S4, S5 Table), with at least 16.4% of the genomic windows assigned

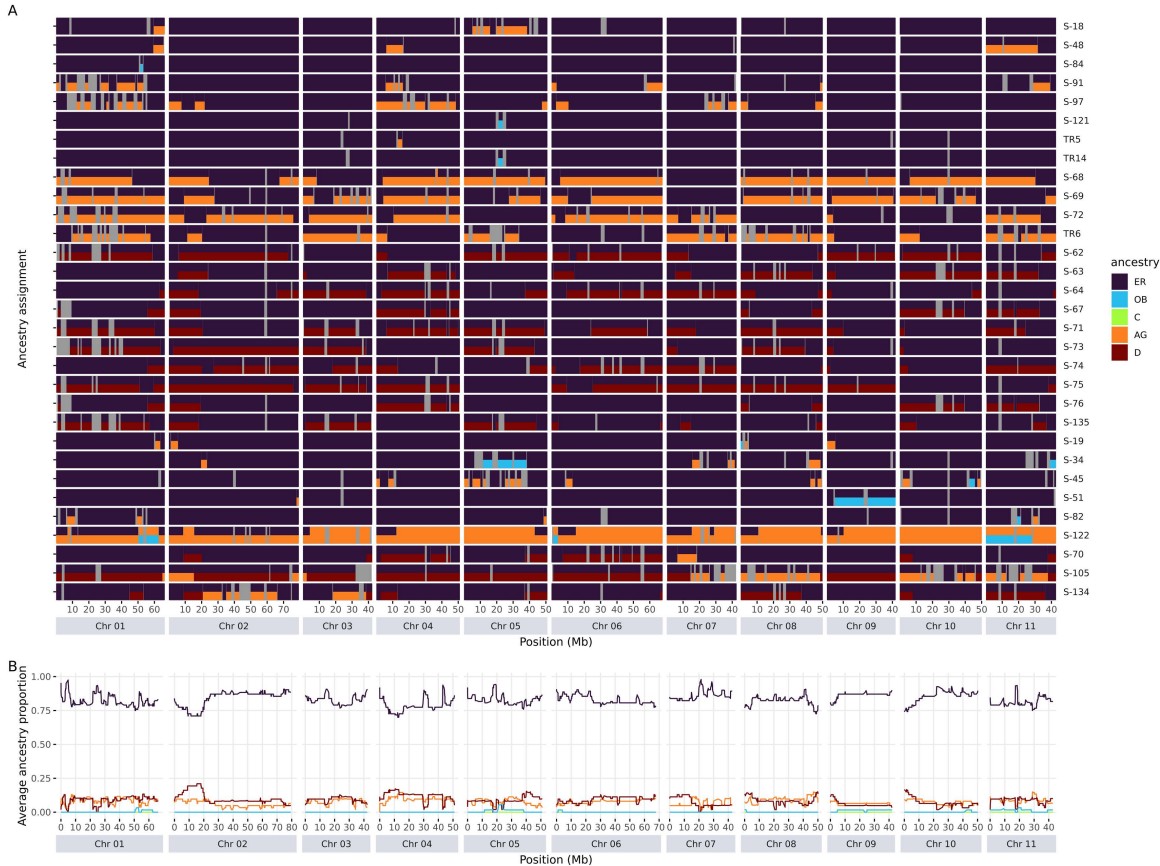

**Fig 3. Genome-wide introgression in 31 admixed individuals from the core set.** (A) Local ancestry proportion (0, 50%, 100%) inferred along the chromosomes. The ancestry groups are highlighted by colors (AG – orange, OB – blue, D – red, ER – dark purple), and undetermined regions are in gray. (B) Average ancestry inference across the admixed individuals at each SNP position. The remaining 14 accessions of the core set were estimated to only have ER ancestry.

to ER. Different admixture patterns and tract lengths were detected in the 45 core set accessions. Fourteen were exclusively of ER origin, while 31 individuals were detected as admixed. Among the two-way admixed genotypes, eight individuals showed < 10% admixture from the AG or OB groups, four individuals had AG segments representing > 20% of the genome, while 10 individuals had segments from the D group > 10%. Nine individuals showed three-way admixture, i.e., either ER x AG x OB or ER x D x AG (Fig 3A). The admixture block lengths ranged from 1 to 76.7 Mb (S5 Fig), with a median at 5.4 Mb for the AG ancestry, and 9.8 Mb for the D ancestry.

We assessed the distribution pattern of the admixed segments throughout the genome by estimating the average ancestry proportions amongst all of the 31 hybrids at each SNP (Fig 3B). Despite the overall varied admixture patterns and segment lengths, the AG and D ancestries had almost equal contributions at the SNP level and were evenly distributed throughout the genome (Fig 3B). In some chromosomes, the admixture level was slightly higher at the beginning or end than in the other regions, such as in chromosomes 2, 9, and 10. By alignment to the recombination map, we observed that elevated admixture levels were in regions with high recombination rates (data from Mérot-L'Anthoëne et al. [7]; available in MoccaDB, https://moccadb.ird.fr/] and Brazier & Glémin [76; Supporting Information]; S6 Fig).

## 4. Discussion

### 4.1. Genetic origins of Vietnamese Robusta coffee

In this study, we used genome-wide SNP data to assess the extent and source of *C. canephora* genetic diversity that has been transmitted from Africa to Vietnam. Our development of the 261 SNP set allowed us to rapidly collect data from different genotyping resources, effectively assess the genetic diversity (origin), and identify clones or redundant individuals with the aim of selecting core accessions. Whole-genome sequencing has facilitated accurate detection of admixture tracts throughout the genome [77], which helped us better understand the hybridization origins and should provide a basis to detect associations with beneficial traits in Vietnamese Robusta coffee.

Wild Robusta individuals from Central Africa were classified differently when we used different genotype data (KASPar genotyping and whole-genome re-sequencing), different sets of SNP markers (261 SNPs and whole-genome SNPs) and different analyses (PCA and sNMF). As there is less genetic differentiation between the E and R, A and G, and O and B groups [7], and here we analysed an unequal number of samples, they were thus merged into the same clusters (ER, AG and OB, respectively). However, they were all congruent with the findings of previous studies [7,10,11], and did not affect the ancestry assignment of the Vietnamese accessions.

Of the previously classified genetic groups, the E and R groups, native to DRC were abundantly represented in the 126 Vietnamese Robusta accessions. However, the Vietnamese individuals were likely not inbred, as suggested by the low inbreeding coefficient, they were also related to wild ER accessions from different sources within the Congo Basin (Sankuru, Yangambi, or Epulu) (S1 Appendix). This finding is consistent with previous Vietnamese genotyping studies [59,60], although here we report more precise source origins. Our findings confirmed historical records of the early diffusion of Robusta (Congolese) types, under the name "Robusta", from Sankuru (DRC) to Southeast Asia around 1900 via the Java Coffee Research Station, Indonesia — the first major Robusta breeding and worldwide distribution center [14,22,78]. Subsequent introductions from DRC (1930–1960) mostly involved elite hybrids developed at the Lula and Yangambi research stations and widely distributed by the Institut National pour l'Étude Agronomique du Congo belge (INEAC [22]) (S1 Appendix).

The Congolese material disseminated to Vietnam was likely introduced through various pathways and in multiple waves in the form of wild, traditional, or elite clones. Among the selected elite varieties commonly used as maternal coffee breeding materials in Vietnam, some had been imported from Indonesia [58]. Moreover, in the 1980s, Vietnamese farmers were sent to Indonesia by the Vietnamese government to study coffee cultivation, which may have led to some material exchanges from the Coffee and Cocoa Research Institute (ICCRI) breeding center or plantations in Lampung province

(Ucu Sumirat, personal communication). There are also reports mentioning that varieties from DRC were initially introduced to the Central Highlands [56]. Specifically, our eleven historical accessions from the Dak Lak Museum collection, which might be the earliest introduced varieties, and all but one of the elite TR clones, had an 86–100% ancestral proportion from the ER group. This indicates that the Robusta coffee cultivated in Vietnam primarily originates from Congo Basin genetic resources.

However, Congolese populations are not the only wild sources. In at least 31 accessions, we found traces of all other genetic groups, except group C (which included individuals from northwestern Congo, southeastern Cameroon, and southwestern CAR [5]). The two major admixed sources in the hybrid accessions of the WASI core set were from the D (Guinean group) and AG (Kouilou-Angola group) groups, and to a lesser extent the OB group (East CAR-Uganda origin). Two historical accessions represented 10% and 12% of the OB group, in line with Javanese historical records. When Robusta seeds from Congo were initially sent to Java under the name "Robusta" (*Coffea robusta* L.Linden), it was quickly accepted by farmers because of its high productivity and apparent resistance to coffee leaf rust [14]. These materials were subsequently enriched with other sources such as Sankuru-DRC (*Coffea canephora* f. *sankuruensis* De Wild) for the R group, "Kouilou" or "Kwilu" varieties (*Coffea canephora* var. *kouilouensis* Pierre) for the A group, and Uganda (*Coffea ugandae* Cramer) for the O group [14,79]. Admixtures between these origins may have occurred in Vietnam, although hybrid material may also have been introduced directly from Indonesia. Indeed, clones derived from ER x Kouilou hybrids were grown in the Lampung region of Indonesia.

## 4.2. Tapping the genetic diversity for coffee breeding

These different introgressed origins were detected in almost every part of the genome with a Congolese genomic background. As shown by the average ancestry proportion (Fig 3B), the admixture probability seemed to be even throughout the genome, and slightly higher at the ends of some chromosomes, which could have been the result of high recombination rates [7,76]. Moreover, by using local ancestry analysis, which has higher resolution at whole genome level, we identified new haplotype segments from group OB in some individuals (i.e., S-122) that could not be detected by the global admixture analysis. The admixture tracts were distributed differently in each individual, but all of the patterns suggested backcrossing. Hybrids with longer admixture segments might have been the result of recent hybridization, and conversely, those with smaller admixture segments might have undergone more backcrossing cycles, probably before being introduced into Vietnam from Java. Further analysis on local ancestry inference at the haplotype level would be required to precisely estimate their admixture time [80].

The different genetic groups present in the hybrids have been considered complementary in terms of agronomic traits, so they might be useful for breeding [18,81,82]. Moreover, our results suggest that introgression of various fragment sizes with potentially favorable traits could be possible in Robusta within a few backcross generations, since we observed admixture patterns of various sizes evenly distributed throughout the genome. A local ancestry inference distribution and genetic recombination rate map could provide useful background information for introgression breeding [83]. For example, varieties hosting ancestry segments of sought-after genes could be targeted, and potential outcomes could be predicted if the recombination rates are known [84]. In addition, as unwanted alleles are often introgressed along with the targeted ones due to genetic linkage [83], varieties with smaller admixed segments may have lower risk of transmission of undesirable traits. The fact that we already have different admixture combinations within our core-set accessions could facilitate and shorten breeding schemes when selecting the most suitable clones as parental lines for crosses [83].

Most breeding programs have mainly relied on heterosis resulting from crossing Congolese and Guinean groups to improve coffee quality in terms of bean size, disease resistance, etc. [18,82,85,86]. In WASI, there have been attempts to select and produce new varieties with higher performance [58]. Eleven elite varieties have been recognized, which have higher yield and bean quality, with tolerance to late irrigation timing. In the present study, ten of these elite varieties were included in the germplasm collection and the core collection. Seven of them were found to only be composed of

the ER group and the three others had an ER group genomic background with little admixture: two presented only minor proportions of the AG or OB groups, and only one had about 20% admixture from the AG group. This could be explained by the limited genetic resources of the complementary groups. Another possible reason is that, compared to the other groups, the Congolese ER varieties might be better suited to the growing conditions that prevail in the Vietnamese Central Highlands as they are more resistant to leaf rust disease and have a late maturation time adapted to the dry season conditions [58,87]. The only elite variety with significant admixture from the Kouilou-Angola AG group was found to benefit from the late ripening time and leaf rust resistance of the Congolese ER group, but it had lower yield than the other elite clones [58]. The genomic regions associated with yield [88] might be introgressed by the AG group in this variety. We proposed a core set representative of the genetic diversity held in the WASI collection, including hybrids of varying genetic backgrounds. Phenotyping of these core accessions may lead to the identification of beneficial traits to be used in breeding [51].

In the future, Vietnamese Robusta coffee will require increased resilience to cope with climate change and associated abiotic and biotic stresses, which are expected to have a significant impact on the Vietnamese Central Highlands [2,61,89]. Accessions with admixture segments of Kouilou-Angola or Guinean origin have been found to be more drought tolerant and resistant to pests and diseases [18,90,91], and they could thereby help improve coffee adaptation in this setting. They could potentially become more superior varieties, or be used as breeding material to introduce adaptive traits into the elite varieties. Greater insight into the coffee genetics-climate relationship [10,92] might also help to optimize breeding material choices. Moreover, breeding strategies should also be focused on conserving or introducing more diversity from different genetic sources so as to be able to explore new beneficial genetic markers.

## Supporting information

**S1 Table. Passport data of 126 Vietnamese accessions.**
(XLSX)

**S2 Table. Passport data of 127 African reference accessions.**
(XLSX)

**S3 Table. Information of 261 Kaspar SNPs.**
(XLSX)

**S4 Table. Ancestry proportion of 45 core accessions (estimated by sNMF).**
(XLSX)

**S5 Table. Genotyping data of all accessions (253) for the 261 SNP loci, determined from Kaspar genotyping, 8.5k SNP array, and sequencing data.**
(XLSX)

**S1 Fig. Variant calling workflow according to GATK Best Practices and SNP filtering.** Software used: BWA mem 0.7.8, Picard Tools 2.26.9, SAMtools 1.14, GATK 4.2.4.0, vcftools 0.1.16.
(PNG)

**S2 Fig. Cross-entropy criterion of sNMF runs on all individuals with different K values.**
(PNG)

**S3 Fig. PCA results for the 126 WASI Vietnamese germplasm accessions and African genetic group reference set, using 261 SNPs.**
(PNG)

**S4 Fig. Genetic structure of the African reference set individuals based on sequencing data.** Five ancestral groups were estimated from the best sNMF run using a subset of 5 kb thinned SNPs.
(PNG)

**S5 Fig. Distribution of admixture block lengths.** The admixture blocks were constituted of continuous admixture patterns. As the local ancestry inference was not based on phased sequencing data, the exact ancestry segments were unknown, but based on the parsimony principle, the probability of having 2 crossover events is higher than 3 or more events. The admixture blocks defined here were therefore with the most likely lengths.
(PNG)

**S6 Fig. Distribution of the genetic groups and recombination rates throughout the genome.** (A) Average proportion of the genetic groups at each SNP position. (B) Genetic distances along the chromosomes obtained from Mérot-Anthoëne et al. [7].
(PNG)

**S1 Appendix. Focus on the contribution of central Congolese (OB-ER) genetic groups to Vietnamese coffee trees.**
(PDF)

## Acknowledgments

The authors acknowledge the ISO 9001 certified IRD i-Trop HPC (South Green Platform) at IRD Montpellier (France) for providing the HPC resources that contributed to the research results reported in this paper. URL: https://bioinfo.ird.fr/-http://www.southgreen.fr.

## Author contributions

**Conceptualization:** Philippe Cubry, Pierre Marraccini, Ngan Giang Khong, Valerie Poncet.

**Data curation:** Thi Nhu Le, Viet Ha Phan, Thi Tieu Oanh Dinh, Thi Bich Ngoc Tran, Van Toan Nguyen, Jean-Léon Kambale, Gba Kossia Manzan Karine, Pascal Musoli, Ucu Sumirat, Jose Cassule Mahinga, Piet Stoffelen, Claude Patrick Millet, Dapeng Zhang, Pierre Marraccini, Valerie Poncet.

**Formal analysis:** Tram Vi, Claude Patrick Millet.

**Methodology:** Tram Vi, Philippe Cubry, Yves Vigouroux, Ngan Giang Khong, Valerie Poncet.

**Writing – original draft:** Tram Vi.

**Writing – review & editing:** Tram Vi, Philippe Cubry, Ucu Sumirat, Piet Stoffelen, Claude Patrick Millet, Pierre Marraccini, Yves Vigouroux, Ngan Giang Khong, Valerie Poncet.

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
