## [Decision Letter · Decision Letter 0]

6 May 2025

Out of Africa: the genomic footprints of Vietnamese Robusta coffee

PONE-D-24-58673

Dear Dr. Vi,

We’re pleased to inform you that your manuscript has been judged scientifically suitable for publication and will be formally accepted for publication once it meets all outstanding technical requirements.

Kind regards,

Rodomiro Ortiz, PhD

Academic Editor

PLOS ONE

Additional Editor's comments: As noted by one of the referees, it is a very well thought research with appropriate planning and work done. The materials are relevant for the study and the methods used adequate to address the research objectives. The analysis are sound and interesting results ensued as main research outputs. Furthermore, its reading was easy to follow. Hence, this manuscript is suitable for publication in the journal.

1. Please ensure that your manuscript meets PLOS ONE's style requirements, including those for file naming. The PLOS ONE style templates can be found at https://journals.plos.org/plosone/s/file? id=wjVg/PLOSOne_formatting_sample_main_body.pdf and https://journals.plos.org/plosone/s/file?id=ba62/PLOSOne_formatting_sample_title_authors_affiliations.pdf

Reviewers' comments:

Reviewer's Responses to Questions

**Comments to the Author**

1. Is the manuscript technically sound, and do the data support the conclusions?

Reviewer #1: Yes

Reviewer #2: Yes

2. Has the statistical analysis been performed appropriately and rigorously? 

Reviewer #1: Yes

Reviewer #2: Yes

3. Have the authors made all data underlying the findings in their manuscript fully available?

Reviewer #1: Yes

Reviewer #2: Yes

4. Is the manuscript presented in an intelligible fashion and written in standard English?

Reviewer #1: Yes

Reviewer #2: Yes

5. Review Comments to the Author

Reviewer #1: This is important, basic genomics work reporting the genomic characterization of economically important coffee germplasm. The paper is very well-written, the methods and interpretations are sound, and I find it acceptable for publication as is.

Reviewer #2: The manuscript entitled “Out of Africa: the genomic footprints of Vietnamese Robusta coffee” provides a genetic diversity study of 126 Robusta accessions from the Vietnam coffee germplasm collection together with reference samples from previously characterized diversity groups of Africa. The paper is well written, and the analyses provided are sound and well conducted. Resulting data are available for the scientific community and might contribute to further studies on Robusta genetic diversity worldwide. More importantly, climate change threatens coffee cultivation around the world and genomic contributions for more precise breeding strategies are urgently needed. In that sense, we recommend the publication of the submitted paper despite the fact that was a very descriptive work.

6. PLOS authors have the option to publish the peer review history of their article (what does this mean? ). If published, this will include your full peer review and any attached files.

**Do you want your identity to be public for this peer review?** For information about this choice, including consent withdrawal, please see our Privacy Policy .

Reviewer #1: **Yes: ** Melinda Yerka

Reviewer #2: No

---

## [Editor Report · Acceptance letter]

PONE-D-24-58673

PLOS ONE

Dear Dr. Vi,

I'm pleased to inform you that your manuscript has been deemed suitable for publication in PLOS ONE. Congratulations! Your manuscript is now being handed over to our production team.

Kind regards,

on behalf of

Professor Rodomiro Ortiz

Academic Editor

PLOS ONE